# Impact of COVID-19 and Antibiotic Treatments on Gut Microbiome: A Role for *Enterococcus* spp.

**DOI:** 10.3390/biomedicines10112786

**Published:** 2022-11-02

**Authors:** Elda Righi, Lorenza Lambertenghi, Anna Gorska, Concetta Sciammarella, Federico Ivaldi, Massimo Mirandola, Assunta Sartor, Evelina Tacconelli

**Affiliations:** 1Infectious Diseases Unit, Department of Diagnostics and Public Health, University of Verona, 37134 Verona, Italy; 2Microbiology Unit, Udine University Hospital, 33100 Udine, Italy

**Keywords:** COVID-19, gut dysbiosis, gut microbiome, *Enterococcus*, antibiotics

## Abstract

Objective: Several studies showed the substantial use of antibiotics and increased risk of antimicrobial resistant infections in patients with COVID-19. The impact of COVID-19-related treatments and antibiotics on gut dysbiosis has not been clarified. Design: The prospective cohort study included hospitalized COVID-19 patients (April–December 2020). The gut microbiome composition was analysed by 16S sequencing. The gut diversity and changes in opportunistic bacteria (OBs) or symbionts were analysed according to clinical parameters, laboratory markers of disease progression, type of non-antibiotic COVID-19 treatments (NACT) and type, WHO AWaRe group, and duration of antibiotic therapy (AT). Results: A total of 82 patients (mean age 66 ± 13 years, 70% males) were enrolled. The relative abundance of *Enterococcus* was significantly correlated with duration of hospitalization, intensive care unit stay, O_2_ needs, and D-dimer, ferritin, and IL-6 blood levels. The presence of *Enterococcus* showed the highest number of correlations with NACT, AT, and AT + NACT (e.g., hydroxychloroquine ± lopinavir/ritonavir) and increased relative abundance with AWaRe Watch/Reserve antibiotics, AT duration, and combinations. Abundance of *Dorea*, *Agathobacter*, *Roseburia*, and *Barnesiella* was negatively correlated with AT and corticosteroids use. Patients with increased IL-6, D-dimer, and ferritin levels receiving AT were more likely to show dysbiosis with increased abundance of *Enterococcus* and *Bilophila* bacteria and decreased abundance of *Roseburia* compared with those not receiving AT. Conclusion: Microbiome diversity is affected by COVID-19 severity. In this context, antibiotic treatment may shift the gut microbiome composition towards OBs, particularly *Enterococcus*. The impact of treatment-driven dysbiosis on OBs infections and long-term consequences needs further study to define the role of gut homeostasis in COVID-19 recovery and inform targeted interventions.

## 1. Introduction

COVID-19 has been associated with gut microbiome dysbiosis, causing an increase in the rate of opportunistic bacteria (OBs) to the detriment of beneficial symbionts [1,2,3]. Frequently reported OBs in association with SARS-CoV-2 infection include, but are not limited to, species of *Enterococcus*, *Bilophila*, and *Rothia*, and various species of Enterobacterales [1,3,4,5,6,7,8]. On the other hand, bacteria usually involved in the prevention of intestinal inflammation and homeostasis regulation such as species of *Roseburia* or in the Ruminococcaceae family exhibited significant changes in the gut of infected patients, resulting in overall decreased abundance [1,6,9,10]. Changes in beta diversity were also reported, showing significant differences in Bray–Curtis index among SARS-CoV-2 infected vs. noninfected patients [1,9].

Most studies have focused on intestinal dysbiosis and disease severity or compared gut microbiome profiles between COVID-19 and other viral diseases [9,10], while reports investigating the impact of antibiotics or COVID-19 treatments are lacking. The majority of those hospitalized with COVID-19 receive antivirals, antibiotics, and/or inflammatory drugs that may affect the gut composition while treating the disease or other associated infections. Moreover, empirical antibiotic therapy has been reported in up to 75% of hospitalized patients and far exceeds the estimated rate of bacterial coinfection that is set around 9% [11]. Since the beginning of the pandemic, the potential overuse of antibiotics has been concerning due to its potential in the selection of OBs and antibiotic resistance that may complicate the course of the disease and limit treatment options [12]. Antibiotics are often used in association with oxygen (O_2_) therapy as well as exploratory treatments that have now been discontinued (e.g., hydroxychloroquine, lopinavir/ritonavir) or more recently established COVID-19 treatments (e.g., tocilizumab, remdesivir, corticosteroids, etc.) [13]. 

We have analysed the gut microbiome changes in a cohort of COVID-19 hospitalized patients exploring the correlations between abundance of symbiont or opportunistic genera with clinical, laboratory, and therapy data.

## 2. Methods

### 2.1. Study Population 

Inclusion criteria were 1. a documented SARS-CoV-2 infection requiring hospitalization and 2. age ≥ 18 years. Stool and plasma samples were collected at the same time and within the first 2 weeks of hospitalization (median hospitalization day 4, Q1–Q3 2–9). If more than one sample was collected for each included patient, only the first one was considered in the analysis. 

Data included at the time of the sample’s collection included age, comorbidities, SARS-CoV-2 serology, laboratory parameters, need and level of oxygen (O_2_) support, and type of treatment. Age was used as a continuous variable and reported according to three age groups, considering as cut-off points 50 and 65 years of age due to an increased COVID-19 severity according to age reported in the literature [14]. The type of O_2_ support during hospitalization was reported as no O_2_ needed (room air), low O_2_ support (e.g., nasal cannulae, face masks, or reservoir masks), or high O_2_ support (either non-invasive ventilation or mechanical ventilation). The fraction of inspired oxygen (FiO_2_), indicating the need for supplemental O_2_ therapy), was also collected. Regarding treatment, any non-antibiotic COVID-19-related treatment (NACT) that was available at the time of data collection (e.g., hydroxychloroquine, lopinavir/ritonavir, remdesivir, tocilizumab, corticosteroids, heparin) was included in the analysis. Antibiotic treatment (AT) was reported according to two factors: 1. The class of AT, including beta-lactams (amoxicillin/clavulanate, piperacillin/tazobactam, ceftriaxone, and meropenem), anti-intracellular bacteria antibiotics (ciprofloxacin, doxycycline, clarithromycin, and azithromycin), or anti-Gram-positive bacteria (GBP) agents (vancomycin, daptomycin, and linezolid) and 2. The 2021 WHO AWaRe classification (e.g., Access, Watch, and Reserve antibiotics) that groups AT according to their resistance potential [15]. Antibiotic combination was defined as the association of two or more antibiotics. The combinations of AT and NACT were also reported. To account for the impact of treatment on gut microbiome, we considered in the analysis any treatment that was administered for at least 24 h and that was started at least 48 h before stool collection but discontinued no more than 7 days before stool collection. 

The study protocol was approved by the institutional review board at Verona University Hospital (CO-BIOME study, IRB number 2906). Informed consent was provided by all study participants. 

### 2.2. Laboratory Analyses 

The SARS-CoV-2 diagnosis was performed by nasopharyngeal swab using real time multiplex reverse transcription polymerase chain reaction (multiplex RT-PCR) for simultaneous detection of three different viral targets (E, N and RdRP genes) from nasopharyngeal and oropharyngeal swabs using the Allplex 2019-nCoV assay kit (Seegene, Seoul). Laboratory parameters that were available at the time of stool collection are reported in the analysis. The detection of the SARS-CoV-2 antibody response was performed using the Elecsys^®^ Anti-SARS-CoV-2 ECLIA assay (Roche Diagnostics AG, Rotkreuz, Switzerland) measuring anti-S with threshold for positivity of 0.8 WHO binding antibody units (BAU)/mL. Antibody titers were reported as negative or borderline (≤5 BAU/mL), low to moderate (5 < BAU/mL < 100), or high (≥100 BAU/mL). Markers of systemic inflammation such as blood levels of ferritin (ng/mL), D-dimer (ng/mL), C-reactive protein (CRP, mg/L), and neutrophil/lymphocyte ratio (NLR) [16] were collected. The cytokine EDTA plasma levels were analysed by using enzyme-linked immunosorbent assay (ELISA) according to the manufacturer’s instructions (BioLegend, Milan, Italy) and included IL-6 (range 7.8–500 pg/mL), IL-10 (range, 3.9–2500 pg/mL), IL-2 (range 7.8–500 pg/mL), INF-gamma (INF-g, range 7.8–500 pg/mL), TNF-alpha (TNF-a, range 7.8–500 pg/mL), IL-17A (range, 3.9–250 pg/mL). IL-6 levels, that have been previously associated with severe COVID-19 were stratified as normal range (<15 pg/mL), moderately increased (15–49 pg/mL), and high (>50 pg/mL) [17].

To perform 16S rRNA sequencing of stool samples, the collected samples were preserved at −80 °C until nucleic acids extraction. Nucleic acids were isolated with the Norgen Stool DNA Isolation Kit (Norgen Biotek, Ontario, Canada) following manufacturer’s instructions. The DNA samples were subsequently quantified using the Maestrogen MN-931A MaestroNano Pro spectrophotometer (Maestrogen, Brumath Cedex, France) and normalized to a concentration of 50 ng/μL, then stored at −80 °C. The 16S rRNA gene V1-V3 region was amplified using previously defined primers and underwent paired-end sequencing using version 3 (300 × 2) chemistry on the MiSeq instrument (Illumina) according to manufacturer’s instructions [18]. 

### 2.3. Data Analysis

The quality of reads was controlled using FastQC. The pre-processing, trimming, filtering, and merging was carried out with Cutadapt [19]. Reads were subsequently aligned against the SILVA ARB 16S rRNA v. SSU 138 database [20] with minimap2 [21]. Next, the taxonomic profiles were computed using MEGAN6 [22] and exported into CVS files on multiple taxonomic levels in a summarized manner. The downstream analysis and visualizations were carried out in Python 3.8 with matplotlib v. 3.2.1 [23] and scipy [24]. Computation of the weighted UniFrac [25] was carried out in Python 3.8 using a scikit-bio [26] and ete3 [27] software packages. Beta diversity metrics and relative abundance of genera ≥0.1% were reported. Median and ranges were used for continuous variables, count and percentages for nominal variables. Spearman plots was used to report the correlations between relative abundance of bacteria (at genus and family levels, shown in alphabetical order) and clinical, laboratory, and treatment data at the time of sample collection. Only bacteria detected in at least 20 patients were shown in the plots. Treatment data were reported as number of days receiving AT and/or NACT (except for tocilizumab, that is usually administered as single infusion due to its long half-life [28]. Each treatment or treatment combination that was ongoing at the time of stool collection was considered independently. The Kruskal–Wallis test was used for comparing bacterial abundance of independent groups that were reported as bar charts. Only bacteria with significantly different abundance were reported in the bar charts apart from those in *Alistipes* and *Pseudomonas* (due the overall high abundance of *Alistipes* and the potential clinical significance of *Pseudomonas*). The association between categorical variables was assessed using Fisher’s test. To search for potential confounders (age, gender, length of hospital stay, O_2_ support, IL-6, ferritin, and D-dimer levels) that may have predicted AT use, a logistic regression was performed. Stata Version 16.1 (College Station, TX, USA: StataCorp LP) was used, considering a two-tailed α error of 0.05 in all analyses. 

## 3. Results

A total of 116 consecutive patients hospitalized with COVID-19 between April 1st and December 31st, 2020, were screened for inclusion at a tertiary referral hospital in Italy. Of these, 20 with uncomplete data and 14 who did not sign informed consent were excluded, leaving 82 patients that were included in the study. Mean age was 66 ± 13 years and 70% of patients were males. Patient demographic and clinical characteristics are summarized in Table 1. A third of COVID-19 patients reported no comorbidities while, among those with comorbidities, 87% suffered from hypertension. Serology was positive in 74% of patients at the time of stool collection. Most patients (83%) necessitated O_2_ therapy. Eighteen (22%) patients required ICU admission and seven (9%) died during hospitalization. Patients with disease progression requiring ICU admission compared to those not requiring intensive care more frequently had ferritin > 400 ng/mL (*p* = 0.02), D-Dimer > 500 ng/mL (*p* < 0.001), and IL-6 > 15 pg/mL (*p* = 0.02). Microbiologically documented concomitant infections were reported in 15 (18%) patients and are described in Appendix A.

### 3.1. Gut Microbiome and Clinical Markers

Correlations between the relative bacterial abundance and clinical parameters are reported in Figure 1. Age did not show significant correlations with OBs or symbionts. When grouped by age categories, we did not see differences in the distribution of *Bacteroidetes* and *Firmicutes*, while the Jaccard dissimilarity was significantly different between patients below 50 years and those aged 65 or older (*p* = 0.03), but a clear pattern in favouring OBs according to age was not shown (Appendix A). The length of hospitalization and ICU stay positively correlated with Enterococcaceae and *Enterococcus* (at family and genus levels) and negatively correlated with Lachnospiraceae and butyrate-producing beneficial symbionts such as *Agathobacter* and *Roseburia*. No differences in gut microbiome characteristics were noted according to SARS-CoV-2 serological results (Appendix A). Markers of systemic inflammation and COVID-19 progression showed positive association with OBs genera and negative correlation with beneficial symbionts. *Enterococcus* showed positive correlation with increased ferritin and D-dimer and *Pseudomonas* with CRP levels. *Agathobacter* was negatively correlated with ferritin and NLR. Regarding cytokines, at a genus level, negative correlations were observed for *Pseudomonas* with IL-6, TNF-a, IL-17A, and INF-g. Positive correlations were shown for OBs such as *Coprobacter* and *Desulfovibrio* and Il-6 and TNF-a, between *Enterobacter* and IL-10, and between *Pseudomonas* and IL-2 levels. At a family level, proinflammatory cytokines such as IL-6 and TNF-a had positive correlation with Barnesiellaceae, Sulfovibrionaceae, and Tannerellaceae. 

The distribution of bacterial abundance was tested according to normal, moderately increased, and high levels of the key cytokine IL-6, showing significantly increased abundance of *Firmicutes* and *Enterococcus* in COVID-19 patients with high vs. normal IL-6 levels (*p* = 0.05 and *p* = 0.02) and between high vs. moderate and normal IL-6 levels (*p* = 0.04), respectively (Figure 2). No differences in beta diversity were noted. 

### 3.2. Gut Microbiome and O_2_ Therapy

Figure 1 showed a negative correlation between FiO_2_ and the abundance of pathobionts such as Lachospiraceae and *Agathobacter* at a family and genus levels, respectively. Significant differences in the distribution of pathogens’ abundance at a genus level according to the level of O_2_ support are reported in Figure 3. Significantly increased abundance of *Enterococcus* and lower relative abundance of *Roseburia* were observed among patients receiving high compared to low and no O_2_ support (*p* = 0.04 and *p* < 0.001 *Enterococcus*; *p* = 0.04 and *p* = 0.01 *Roseburia*), while higher relative abundance of *Paraprevotella* and *Haemophilus* were reported among patients with no O_2_ support compared to those with low O_2_ support (*p* = 0.03 and *p* = 0.05 respectively). The beta diversity of stool samples according to the maximum level of O_2_ support received during COVID-19 hospitalization was analysed (Figure 3), showing a significant difference in the Jaccard similarity index especially in patients receiving low vs. with high O_2_ support (*p* = 0.01). The general profile composition of samples at the phylum level, according to O_2_ support and ordered by relative abundance of Bacteroidetes is reported in Figure 3. No significant differences were shown among groups between the two phyla (Appendix A). The distribution of patients by type of O_2_ support did not significantly differ according to the gender. Among females, only four received high O_2_ support showing increased abundance of *Firmicutes* compared to low O_2_ support (*p* = 0.03), while no significant associations were noted at the genus level (Appendix A), while males, similarly to the overall sample results, showed increased *Enterococcus* (*p* = 0.01) and decreased abundance of *Roseburia* (*p* < 0.001) for high compared with low O_2_ support (Appendix A).

### 3.3. Gut Microbiome and Therapy

Details regarding the type of treatment received at the time of stool collection are summarized in Appendix A. Overall, 15 (18%) patients did not receive either AT nor NACT, 9 (11%) only AT, 40 (49%) a combination of AT + NACT, and 18 (22%) only NACT. 

### 3.4. Antibiotic Therapy (AT)

No differences in the relative abundance of bacterial genera (except for higher abundance of *Roseburia* and *Escherichia* among patients not receiving AT vs. those receiving AT, *p* = 0.01), phyla, and gut microbiome beta diversity were found between patients receiving AT and those not receiving AT (Appendix A). According to the WHO AWaRe classification, 18 (36%), 29 (59%), and 3 (5%) COVID-19 patients were treated with Access, Watch, or Reserve antibiotics, respectively. Bacteria in the phylum Proteobacteria showed increased relative abundance (*p* < 0.001) in patients receiving Watch/Reserve vs. access AT. At a genus level, *Enterococcus* increased relative abundance (*p* = 0.02) and *Blautia* decreased relative abundance (*p* = 0.01) as were noted in the Watch/Reserve compared with the Access group (Figure 4). The Jaccard index was significantly different (*p* = 0.04) between the two treatment groups (Figure 4). No association was found between clinical and laboratory parameters that are indicative of COVID-19 progression (age, gender, length of hospital stay, O_2_ support, IL-6, ferritin, and D-dimer levels) with the use of AT (Appendix A).

Mean time on AT (at the time of stool collection) was 7 days (Q1–Q3, 4–8 days). A negative correlation was observed between the days on AT and abundance of beneficial commensals such as Lachnospiraceae (family level) with a decrease in *Agathobacter*, *Dorea*, and *Roseburia* (genus level); conversely, *Enterococcus* showed positive correlations with AT (in particular, beta-lactams and anti-GPB) duration (Figure 5). The use of combination of two or more antibiotics was associated with a significant increase in abundance of Enterococcaceae and Erysipelotrichaceae and a decrease in Lachnospiraceae. 

### 3.5. Non-Antibiotic COVID-19 Treatment (NACT)

Significant correlations shown by bacterial genera and families with NACT and AT + NACT combinations are reported in Figure 5. *Enterococcus* showed the highest number of correlations with AT, NACT, and AT + NACT, particularly with hydroxychloroquine (HCQ), HCQ and lopinavir/ritonavir (LPV/r), often used in combination in the early phases of the COVID-19 pandemic and subsequently discontinued, and the combination of either HCQ or LPV/r with corticosteroids. Conversely, the current standard-of-care for COVID-19, based on the combination of corticosteroids and heparin, showed a negative correlation with *Enterococcus* and *Barnesiella* and a positive correlation with *Pseudomonas*. Among other treatments, remdesivir in association with AT correlated with *Pseudomonas* and increase in Pseudomonaceae at genus level and with decrease of bacterial families such as Barnesiellaceae, Christensenellaceae, and Lachnospiraceae, while association of remdesivir with corticosteroids or heparin showed positive correlations with the OB *Akkermansia*. Corticosteroids were associated to a significant decrease of beneficial symbionts (*Agathobacter*, *Blautia*, *Dorea*, and *Roseburia*). Patients receiving tocilizumab showed increased relative abundance of *Desulfovibrio* compared with those not treated with tocilizumab (Appendix A).

Figure 6 summarizes the genera and families that have been more frequently associated with AT and/or NACT that are currently used in clinical practice. At the family and genus levels, only Enterococcaceae and *Enterococcus* and Pseudomonaceae and *Pseudomonas* showed positive correlation with AT and with corticosteroids plus heparin, respectively. Several negative correlations were observed between beneficial symbionts and both AT and NACT. Differences in bacterial relative abundance, profile composition and beta-diversity were investigated according to a composite risk profile, including IL-6 > 15.0 pg/mL, D-dimer > 500 ng/mL, ferritin > 400 ng/mL (associated to ICU admission in our study) and use of AT most frequently associated with dysbiosis and that are frequently used in the hospital setting (e.g., beta-lactams and anti-GPB antibiotics). This high-risk group showed increased relative abundance of *Enterococcus* (*p* < 0.001) and *Bilophila* (*p* = 0.03) and decreased abundance of *Roseburia* (*p* = 0.01) compared to patients not showing this profile (low risk). No differences between groups were detected for profile composition at phylum level and beta-diversity.

## 4. Discussion

Our study confirmed the association of COVID-19-related inflammation and severity with an increase in OBs abundance and the reduction in beneficial symbionts in the gut microbiome of hospitalized patients with acute disease. *Agathobacter*, a beneficial symbiont belonging to Clostridiales, showed negative correlation with the fraction of inspired oxygen (FiO2), which is strictly related to the severity of respiratory insufficiency of COVID-19, and with NLR, a predictive marker for critical illness in COVID-19 [16]. Conversely, abundance of *Pseudomonas*, one of the most known OBs, was associated with the increase in inflammatory markers such as CRP and IL-2. The acute-phase protein and active regulator of host innate immunity, CRP, was reported to be predictive of the need for mechanical ventilation in patients with COVID-19-related uncontrolled inflammation [29]. Key proinflammatory cytokines such as TNF-alpha and IL-6 were associated with increased abundance of OBs such as *Coprobacter* and *Desulfovibrio*. Several other cytokines have been investigated in COVID-19 to allow early identification or even prediction of disease progression [30]. Besides IL-16, the anti-inflammatory cytokine IL-10 was found elevated in severe COVID-19 patients [30], while decreased circulating INF-gamma levels were associated to lung fibrosis in COVID-19 patients [31]. We observed a positive correlation between *Enterobacter* and IL-10 and a negative correlation between *Pseudomonas* and INF-gamma levels. 

As most hospitalized patients with COVID-19 receive treatment for COVID-19 or AT (82% in our study), we investigated the potential impact of NACT and AT on gut dysbiosis. Although one might assume that patients with severe disease more frequently receive treatment, in particular AT in case of suspected or documented coinfections, we did not observe a correlation between AT and age, gender, O_2_ therapy, length of stay, or inflammation markers. Gut dysbiosis, however, was documented with any type of NACT and AT. 

The abundance of *Enterococcus* was associated with Watch/Reserve compared to Access AT and was the only genus that positively correlated with any AT and AT combinations. Several beneficial symbionts negatively correlated with AT. Microbiome composition can be rapidly altered by exposure to AT that can also cause immediate collateral damage, for instance through the selection of resistant OB leading to acute disease [32]. Several antibiotic classes, if not all, have been associated with gut dysbiosis. Four-day exposure to beta-lactams and fluoroquinolones showed reduced alpha and beta diversity and increased serum inflammatory cytokines in murine models; furthermore, OB such as *Enterococcus* and *Clostridioides* were significantly enriched in the treated groups, whereas the butyrate-producing bacteria *Blautia*, *Lachnoclostridium*, and *Roseburia* were less abundant [33]. Vancomycin has been associated with reduced proportion of Tregs and Th17, neutrophil-mediated killing, altered expression of bactericidal compounds and with selection of Gram-positive organisms such as vancomycin-resistant enterococci [34,35]. An increase in OBs such as *Desulfovibrio* and *Enterococcus* was significantly correlated with HCQ and LPV/r that have now been discontinued due to a lack of efficacy and increased toxicity [36]. LPV/r is a protease inhibitor used for the treatment of HIV/AIDS mainly in combination with other antiretroviral drugs [37]. Reports on the impact of antiretrovirals on gut microbiome showed reduced abundance of Ruminococcaceae [38] and failure in reversing HIV-induced gut microbiome dysbiosis. High-dose HCQ (100 mg/kg/day) administered for 14 days altered the structure, richness, and the community diversity of the gut microbiota in a murine model and increased the relative abundance of members of the phylum Bacteroidetes while decreasing abundance of members of the phylum Firmicutes compared to controls [39]. Previous data from patients with rheumatologic disease receiving long-term HCQ treatment showed dose-dependent intestinal dysbiosis, and some authors speculated that unfavourable clinical data on HCQ administration in COVID-19 could have been influenced by its impact on gut dysbiosis [40]. We observed limited OBs increase with the use of corticosteroids but a decrease in several beneficial symbionts. Dexamethasone use in mice with inflammatory bowel disease showed that phylum Actinobacteria, and genera *Bifidobacterium* and *Lactobacillus* were significantly increased after treatment. In this study, the potential improvement of inflammation caused by the corticosteroids appeared mediated by a positive effect on the colonic mucin synthesis [41]. In a recent systematic review not including COVID-19, the impact of corticosteroids on the respiratory microbiome was controversial, showing both positive and negative effects and proving that more studies are needed to define their impact [42]. *Akkermansia*, usually found in the mucous layer of the intestine, has been associated with immunomodulatory responses of the intestinal barrier [43] and correlated with days on remdesivir and corticosteroids or heparin. Data on other NACT including remdesivir, however, are scarce, while studies on tocilizumab from patients with active rheumatic arthritis may be biased by disease-specific confounders or concomitant immunosuppressive treatments.

*Enterococcus* can actively traverse the intestinal barrier as can other oxygen-tolerant intestinal pathobionts, such as Enterobacteriaceae, and its abundance has been previously associated with critical COVID-19 disease and with clinically relevant infections such as bloodstream infections [3]. 

A recent systematic review has highlighted the increase in *Enterococcus* in gut microbiome composition in the COVID vs. pre-COVID era and the higher abundance in COVID-19 vs. non-COVID-19 patients [44]. In our study, *Enterococcus* showed a key role both in disease progression and treatment-related dysbiosis. Its abundance was correlated with the highest number of parameters associated with COVID-19 disease progression, including duration of hospitalization, ICU length of stay, D-dimer, and ferritin blood levels. Moreover, the relative abundance of *Enterococcus* was increased in patients with moderate and high IL-6 levels compared with those with normal IL-6 blood levels and among patients with increased O_2_ support compared with those not receiving O_2_ therapy. We also observed an increase in abundance of *Enterococcus* in association with AT, AT combinations, and with HCQ and LPV/r. The only therapy associated with lower abundance of *Enterococcus* was the combination of heparin and corticosteroids, that is now worldwide accepted as the standard of care for COVID-19 treatment [36]. Finally, when we analysed patients at high risk for disease progression and gut dysbiosis, *Enterococcus* was increased in high-risk vs. low-risk patients. 

The main limitations of the study are represented by the lack of post-treatment data, that would increase the understanding of the impact of COVID-19 treatments over time, and by the low number of patients per treatment group that limited the possibility to perform a thorough assessment of each treatment arm. 

In conclusion we have showed that, even if the gut microbiome is affected by COVID-19 irrespective of concomitant treatment, its composition is shifted towards OBs (particularly *Enterococcus*) in hospitalized patients receiving AT during acute disease. Our study supports the use of antimicrobial stewardship protocols to limit the overuse of AT in patients hospitalized with COVID-19. 

Restoring the intestinal homeostasis through pharmacologic means, such as probiotic administration, is currently being explored as a viable treatment option to reshape the gut microbiome in COVID-19 patients [45]. 

Further studies, however, are needed to better understand the potential impact of COVID-19 treatments on gut dysbiosis and the persistence of their effect over time. Specifically, current standard-of-care and potential novel candidate treatments should be investigated along with the potential implications for long-COVID to better define the implications for the future management of COVID-19 hospitalized patients while the evolving situation keeps shaping the guidelines for the management of this disease. 

## Figures and Tables

**Figure 1 biomedicines-10-02786-f001:**
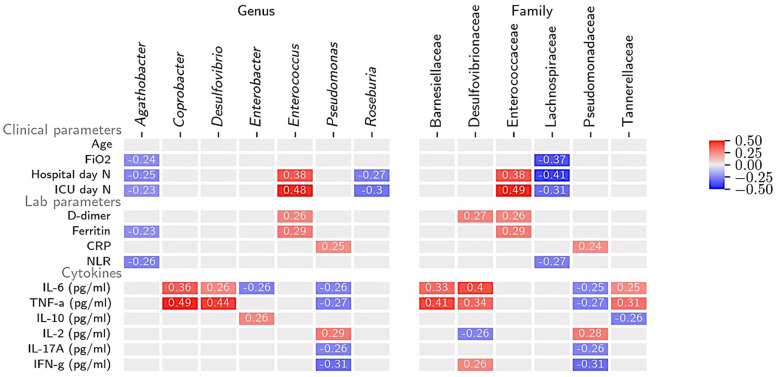
Correlation of gut microbiome composition with clinical and laboratory data at genus and family levels. (1) Clinical parameters. Data associated with COVID-19 progression included age, FiO2, days of hospitalization, and length of ICU stay. A strong positive correlation (>0.35) was observed between *Enterococcus* and duration of hospitalization and ICU days. *Agathobacter* and *Roseburia* and, at family level, Lachnospiraceae showed negative correlations with hospitalization and ICU stay. (2) Laboratory parameters and cytokines. Laboratory data characteristic of COVID-19 progression such as D-dimer and ferritin showed positive correlations with *Enterococcus*. Positive correlation was shown between inflammatory cytokines such as IL-6, TNF alpha with *Coprobacter*, *Desulfovibrio* and between IL-2 and *Pseudomonas*. Bacteria that were detected at least in 20 patients were shown. Only statistically significant (*p* < 0.05) correlations are reported and abundance ≥ 0.1% is shown. The strength of the Rho coefficient is represented by the change in the square colour intensity. Lab = laboratory; FiO_2_ = Fraction of inspired oxygen; ICU = intensive care unit; NLR = neutrophil to lymphocyte ratio; CRP = C-reactive protein. Unit of measurements: D-dimer ng/mL; ferritin ng/mL; CRP mg/dl; IL-6, TNF-alpha (TNF-a), IL-10, IL-2, IL-17A, INF-gamma (IFN-g): ng/mL.

**Figure 2 biomedicines-10-02786-f002:**
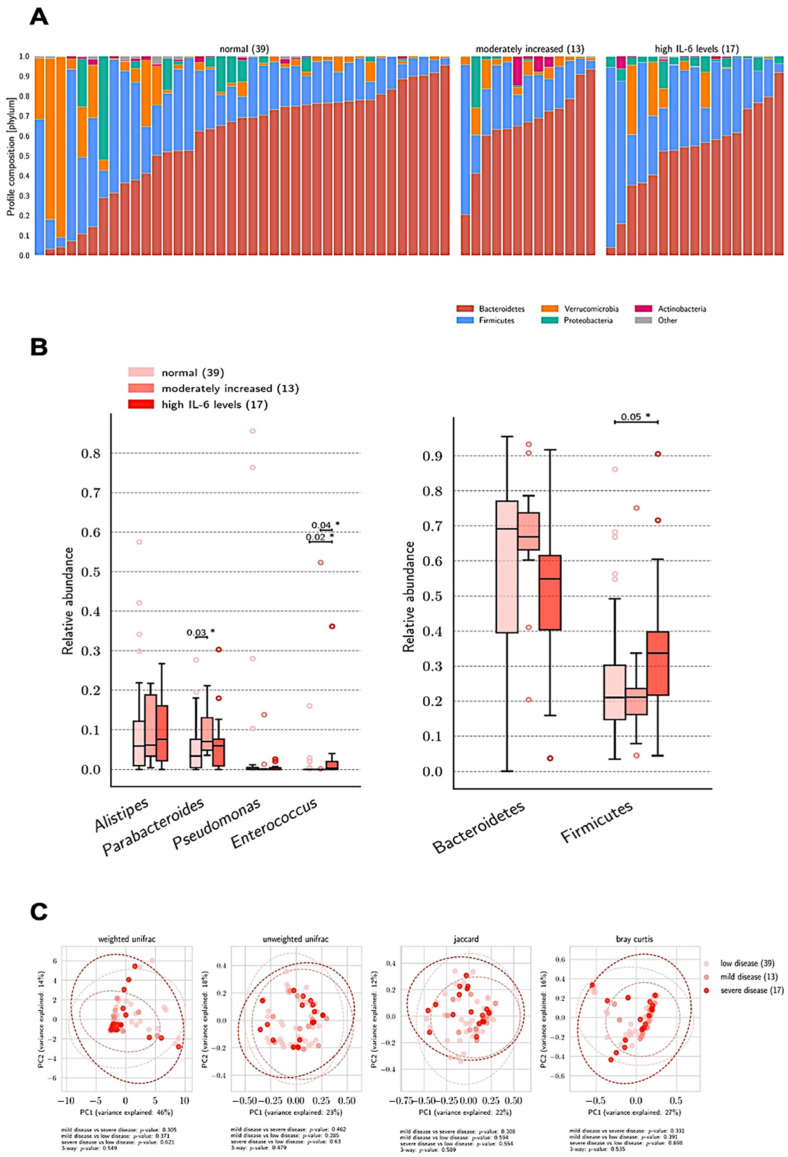
Bacteria relative abundance at phylum and genus levels, beta diversity, and profile composition according to IL-6 levels. (**A**) General profile composition of samples at phylum level was similar among groups. (**B**) Relative abundance analysis did not differ at phylum level but showed higher levels of *Enterococcus* among patients with high IL-6 levels compared with those with moderately increased and normal IL-6. (**C**) PCoA plots of beta diversity between the three groups did not show significant differences. IL-6 measurement was available in 69 (84%) patients. IL-6 levels were reported as normal range (<15 pg/mL), moderately increased (15–59 pg/mL), and high (≥60 pg/mL). Statistical significance reported as * *p* ≤ 0.05.

**Figure 3 biomedicines-10-02786-f003:**
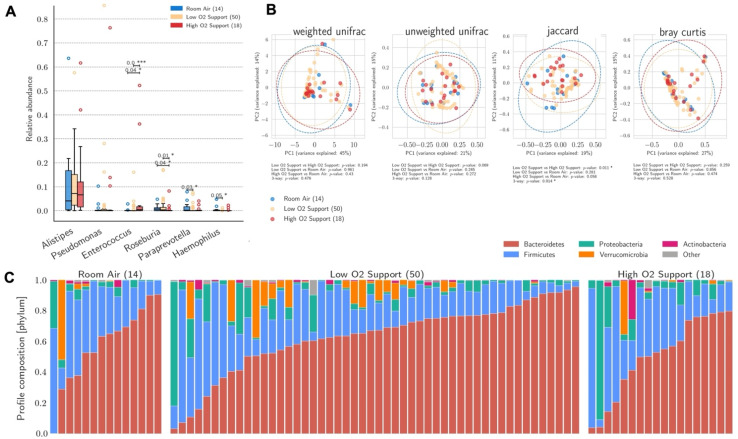
Microbiome profile composition, relative abundance, and bacterial beta diversity according to O_2_ therapy. (**A**) Relative abundance analysis showed higher levels of *Enterococcus* and lower levels of *Roseburia* in patients receiving high compared to low O_2_ support or no O_2_ support (room air). *Paraprevotella* and *Haemophilus* were more abundant in patients with no O_2_ support compared to those with low O_2_ support. (**B**) PCoA plots of beta diversity between the three groups showed a significant difference in richness (Jaccard similarity index) between low and high O_2_ support. (**C**) General profile composition of samples at phylum level in the three O_2_ support groups. Statistical significance reported as * *p* ≤ 0.05, *** *p* ≤ 0.001.

**Figure 4 biomedicines-10-02786-f004:**
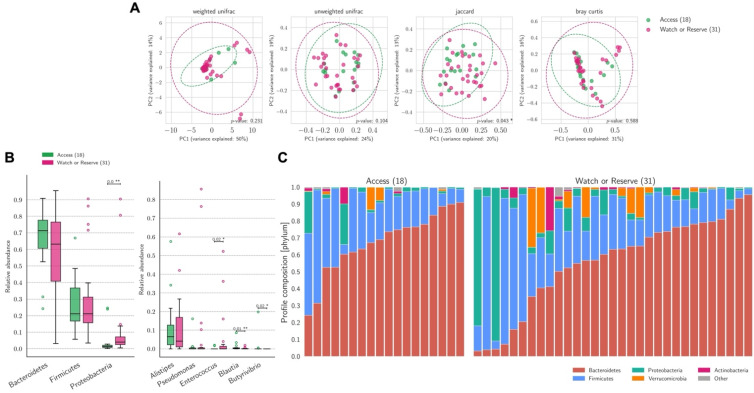
Bacterial beta diversity, relative abundance, and profile composition according to the WHO AWaRe classification for antibiotic use. (**A**) PCoA plots of beta diversity between the three groups showed significant differences in similarity (e.g., Jaccard’s Index) between Access and Watch/Reserve antibiotics. (**B**). Relative abundance analysis showed higher Proteobacteria relative abundance (phylum level) and relative abundance of *Enterococcus* (genus level) for the Watch/Reserve compared to the access group. (**C**) General profile composition of samples at a phylum level. Access antibiotics were compared to pooled Watch and Reserve antibiotics due to the low (*n* = 3) number of treatments using reserve class antibiotics. The 2021 WHO AWaRe classification is available at https://www.who.int/publications/i/item/2021-aware-classification, accessed on 24 September 2022 [15]. Statistical significance reported as * *p* ≤ 0.05, ** *p* ≤ 0.01.

**Figure 5 biomedicines-10-02786-f005:**
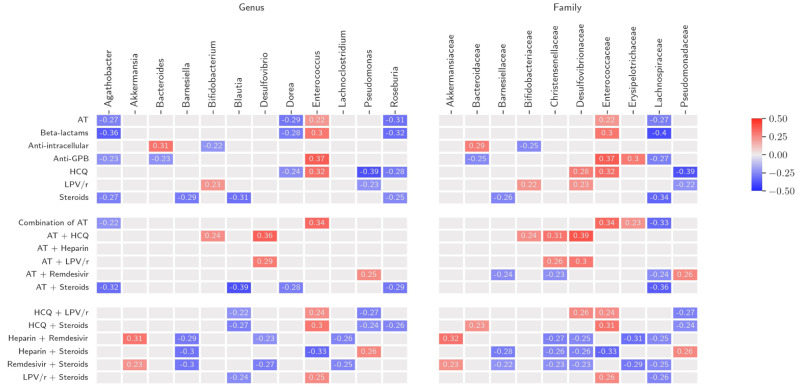
Spearman plot showing the correlation between the gut microbiome composition and treatment duration at the genus and family levels. Lachospiraceae and Barnesiellaceae at family level and symbionts such as *Agathobacter*, *Dorea*, and *Roseburia* at genus level showed negative correlations with COVID-19 and antibiotic treatment. *Enterococcus* showed positive correlations with antibiotic treatment (in particular with beta-lactams and anti-GPB) and HCQ with or without LPV/r and negative correlation only with steroids and heparin that is currently the most commonly used treatment for COVID-19 in hospitalized patients. Families and genera of the bacteria are reported in alphabetical order. Treatment data are shown according to number of days receiving COVID-19 treatment (HCQ, LPV/r, remdesivir, heparin, and steroids) or antibiotic class (beta-lactam, intracellular, or anti-GPB antibiotics) and to the use as a single agent or in combination at the time of sample collection. AT = any antibiotic; GPB = Gram positive bacteria; HCQ = hydroxychloroquine; LPV/r = lopinavir/ritonavir. Beta-lactams included amoxicillin/clavulanate, piperacillin/tazobactam, ceftriaxone, meropenem; intracellular antibiotics included ciprofloxacin, clarithromycin, azithromycin, doxycycline; anti-GPB antibiotics included vancomycin, daptomycin, and linezolid. The strength of the Rho coefficient is represented by the change in the square colour intensity.

**Figure 6 biomedicines-10-02786-f006:**
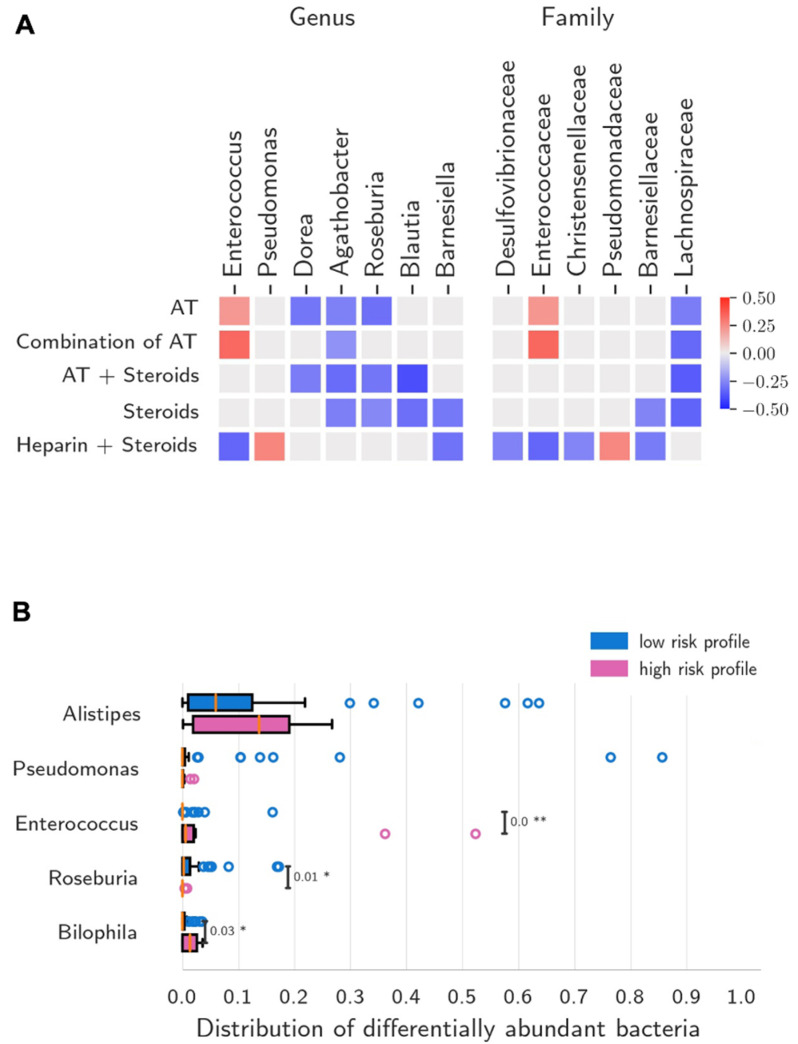
Gut microbiome bacteria associated with treatment and high-risk disease profile. (**A**) Gut microbiome bacteria showing positive or negative correlations with antibiotic therapy (AT) and non-antibiotic COVID-19 therapy (NACT) at genus and family levels. At the genus level, OBs such as *Desulfovibrio*, *Akkermansia* and, most frequently, *Enterococcus*, showed positive correlations with AT and COVID-19 treatments. *Enterococcus* and *Pseudomonas* showed negative and positive correlations with steroids and heparin combination, respectively. *Roseburia*, a beneficial symbiont, only showed negative correlations with treatments. Positive and negative correlations between genus abundance and days on AT and/or NACT are reported in red and blue colours, respectively. AT = any antibiotic treatment; GPB = Gram-positive bacteria; HCQ = hydroxychloroquine; LPV/r = lopinavir/r; steroids = corticosteroids. (**B**) Differences in bacterial relative abundance according to a composite index defining patients at high-risk for disease progression (IL-6 > 15.0 pg/mL, D-dimer > 500 ng/mL, ferritin > 400 ng/mL, and treatment with beta-lactam or anti-GPB antibiotics). The high-risk group showed increased relative abundance of *Enterococcus* and *Bilophila* and decreased abundance of *Roseburia* compared to low-risk patients. Statistical significance reported as * *p* ≤ 0.05, ** *p* ≤ 0.01.

**Table 1 biomedicines-10-02786-t001:** Characteristics of included patients.

Characteristics	COVID-19 Hospitalized Patients (*n* = 82)
Male gender (%)	57 (70)
Median age, years (Q1–Q3)	66 (57–77)
Median hospitalization (H), days (Q1–Q3)	13 (7–22)
Comorbidities (%)	
- No comorbidities	27 (33)
- Hypertension	48 (59)
- Diabetes mellitus	13 (16)
- Heart disease	19 (23)
- Two or more comorbidities	21 (26)
O_2_ Support (%) during H	
- None	14 (17)
- Nasal cannulae or face mask	50 (61)
- Non-invasive or mechanical ventilation	18 (22)
ICU admission (%) during H	18 (22)
Length of ICU stay, days (Q1–Q3)	9 (5–12)
Laboratory parameters (mean ± SE)	
- Ferritin µg/L	944 ± 71
- D-dimer ng/ml	1994 ± 239
- CRP mg/L	47 ± 6.1
- NLR	7.13 ± 0.71

Laboratory parameters were reported at the time of stool collection. ICU = intensive care unit; CRP = C-reactive protein; NLR = neutrophil-to-lymphocyte ratio.

## Data Availability

Available upon reasonable request.

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
