# Peer review of "Impact of COVID-19 and Antibiotic Treatments on Gut Microbiome: A Role for Enterococcus spp."

_biomedicines, 2022, doi:10.3390/biomedicines10112786_

Round 1

Reviewer 1 Report

The authors present the results of the impact of coronavirus infection and antibiotic therapy on the patient's microbiome depending on the severity of the disease and the treatment.

 Unfortunately, I was not able to assess ‘the quality of the presentations’, ‘scientific soundness’, and whether ’the conclusions are supported by the results’, since figures 1-6 are missing from the manuscript.

Author Response

We thanks the reviewer for the comments. We have revised the manuscript (in particular references and methods as suggested) and checked it for potential errors. The figures are embedded in the file that contains the manuscript.

Reviewer 2 Report

The study by Righi et al., “Impact of COVID-19 and Antibiotic Treatments on Gut Microbiome: a role for Enterococcus spp.” reports a cohort study on change in gut diversity during antibiotic and nonantibiotic treatment of patients with COVID-19. The authors demonstrate that even though COVID-19 affects the gut microbiota regardless of the concurrent treatment, its composition is altered towards opportunistic bacteria, notably Enterococcus, during acute disease in COVID-19 hospitalized patients receiving antibiotic therapy. Overall, interesting research on the impact of COVID-19 and antibiotic treatments on the gut microbiome, however, I have the following comments: 

1.      Authors report “Age did not show significant correlations with OBs or symbionts” in line 169, based on the relatively small mean age of 66 ± 13 years. Authors should include wider and different age groups to conclude the age effect on OBs or symbionts.

2.      Authors should include the gender differences in gut dysbiosis even though this study reports 70% male patients. It would be informative and provide a better idea of gender-biased outcomes.

3.      Please correct the merged appearance of the p-value in all the figures including supplementary ones.

4.      The sentence “The use of 2 ore more AT correlated with a significant increase in Enterococcaceae and Erysipelotrichaceae and decrease in Lachnospiraceae.” in lines 227-229, is not making any sense. Authors should reframe it.

Author Response

We thank the reviewer for the comments. We have included the changes in the text (highlighted) and in the figures (replaced or added), as requested. A point-by-point response is provided below.

  1. Authors report “Age did not show significant correlations with OBs or symbionts” in line 169, based on the relatively small mean age of 66 ± 13 years. Authors should include wider and different age groups to conclude the age effect on OBs or symbionts.

Age was used as a continuous variable along with the other parameters in Fig. 1, therefore we believe it was correctly used in the analysis. In response to the reviewer’s comment, we also performed an additional analysis grouping age according to the previous literature on COVID-19 and severity (new CDC reference added in the text), defining 3 age groups (< 50 years old, 50 to 64, and > 64). The results are reported in the text and have been added as a supplementary Figure (Suppl Fig 1). Age was also used as a parameter to adjust for confounders in predicting antibiotic use.

  1. Authors should include the gender differences in gut dysbiosis even though this study reports 70% male patients. It would be informative and provide a better idea of gender-biased outcomes.

We included gender differences according to the disease severity (O2 support) and reported them in the text and added as Supplementary Fig 3B and 3C. Gender was also used as a parameter to adjust for confounders in predicting antibiotic use.

  1. Please correct the merged appearance of the p-value in all the figures including supplementary ones. 

The corrections have been made as requested 

  1. The sentence “The use of 2 ore more AT correlated with a significant increase in Enterococcaceae and Erysipelotrichaceae and decrease in Lachnospiraceae.” in lines 227-229, is not making any sense. Authors should reframe it.

The sentence has been reframed as requested

Round 2

Reviewer 1 Report

I have few minor remarks.

Please check the nomenclatural status of the bacteria mentioned in the text using the LPSN website. If the species, genus, family, etc. are actually published under the ICNP, they must be written in Latin.

Check the bacterial names carefully. For example, I met a typo in the name of genus Bilophila (Line 35).

Check the links carefully. I found one inconsistency; references 20 and 21 should be interchanged (Line 133).

I would advise the authors to indicate in the text in brackets the p-values ​​showing a statistically significant difference between the analyzed groups.

Author Response

We thank the reviewer for the valuable suggestions. Please see the answers below.

Please check the nomenclatural status of the bacteria mentioned in the text using the LPSN website. If the species, genus, family, etc. are actually published under the ICNP, they must be written in Latin.

The bacteria were checked and reported as suggested

Check the bacterial names carefully. For example, I met a typo in the name of genus Bilophila (Line 35).

The names were checked and corrections made accordingly, thank you.

Check the links carefully. I found one inconsistency; references 20 and 21 should be interchanged (Line 133).

The references were corrected.

I would advise the authors to indicate in the text in brackets the p-values ​​showing a statistically significant difference between the analyzed groups.

P values were added where appropriate, thank you.